# Exact and Approximate Conformal Inference for Multi-task Learning

## Abstract

It is common in machine learning to estimate a response $y$ given covariate information $x$. However, these predictions alone do not quantify any uncertainty associated with said predictions. One way to overcome this deficiency is with conformal inference methods, which construct a set containing the unobserved response $y$ with a prescribed probability. Unfortunately, even with a one-dimensional response, conformal inference is computationally expensive despite recent encouraging advances. In this paper, we explore multi-task learning within a regression setting, delivering exact derivations of conformal inference $p$-values when the predictive model can be described as a linear function of $y$. Additionally, we propose `unionCP` and a multivariate extension of `rootCP` as efficient ways of approximating the conformal prediction region for a wide array of multi-task predictors while preserving computational advantages. We also provide both theoretical and empirical evidence of the effectiveness of these methods.

## 1 Introduction

In regression, we aim to predict (or estimate) some response $y$ given covariate information $x$. These predictions alone deliver no information related to the uncertainty associated with the unobserved response, and thus, would benefit from the inclusion of a set $\Gamma^{(\alpha)}(x)$ such that, for any significance level $\alpha \in (0, 1)$,

$$\mathbb{P}\big(y \in \Gamma^{(\alpha)}(x)\big) = 1 - \alpha. \tag{1}$$

One method to generate $\Gamma^{(\alpha)}$ is through conformal inference (used interchangeably with "conformal prediction" in this work) (Gammerman et al., 1998; Lei et al., 2018), which generates *conservative* prediction sets for some unobserved response $y$ under only the assumption of exchangeability. Given a finite number of observations $\mathcal{D}_n = \{(x_i, y_i)\}_{i=1}^n$ and a new unlabelled example $x_{n+1}$, conformal prediction regions are generated through the repeated inversion of the test,

$$H_0 : y_{n+1} = z \quad \text{vs.} \quad H_a : y_{n+1} \neq z, \tag{2}$$

where $z$ is a potential candidate response value, *i.e.,* the null hypothesis (Lei et al., 2018). A $p$-value for this test is constructed by learning a predictive model $\hat{y}(z)$ on the augmented dataset $\mathcal{D}_n \cup (x_{n+1}, z)$ and comparing one's ability to predict the new candidate $z$ using $\hat{y}_{n+1}(z)$ to the already observed responses using, say, $\hat{y}_i(z)$, the predicted value for the $i$-th response as a function of $z$. We note that while $\hat{y}_i(z)$ depends on $D_n$, $x_i$, $x_{n+1}$, and $z$, we only explicitly highlight dependence on $z$. The so-called conformal prediction set is the collection of candidates $z$ for which the null hypothesis is not rejected, *i.e.,* when the error in predicting $z$ is not too high compared to others.

The inversion of the test is Equation (2) is traditionally called "full" conformal prediction since it uses the entire dataset to learn a predictive model. Unfortunately, full conformal prediction is computationally demanding in most cases, with each new candidate point $z$ requiring a new model to be fit. To avoid this complexity, more efficient methods, *e.g.,* split conformal inference (Vovk et al., 2005; Lei et al., 2018) and trimmed conformal inference (Chen et al., 2016), have been introduced with trade-offs between computational efficiency and performance.

Of interest to our work in this paper are *exact* and *approximate conformal inference* methods, which aim to reduce computational complexity without sacrificing performance. Nouretdinov et al. (2001) showed that with certain models, ridge regressors in particular, conformity measures for every observation in a dataset can be constructed as an affine function of the candidate value $z$ and only require training the model once.

In our work, we extend the result of Nouretdinov et al. (2001) to predictors of the form

$$\hat{y} = Hy, \tag{3}$$

where $y$ is an $n \times 1$ vector of responses, $H$ is an $n \times n$ matrix, and $\hat{y}$ is an $n \times 1$ vector of predictions. We note that $H$ can also be a function of a set of covariates, *e.g.,* as with ridge regression where $H = X(X^\top X + \lambda I)^{-1} X^\top$. In reality, the restriction shown in Equation (3) is more general than ridge regression; we only require the predictions be linear functions of the input. In this paper, we refer to models that follow Equation (3) as *linear* models; this is in contrast to the traditional usage of the term to reflect models that are linear with respect to their parameters.

With some models, exact conformal inference is difficult. However, Ndiaye (2022) showed that under certain regularity conditions on the model of interest, we can generate upper and lower bounds on the conformal prediction set, *i.e.,* interval, with only a single model fit, allowing for conservative approximations of the true conformal prediction set. In more complex settings it might be of interest to construct a model for multiple responses, *i.e.,* for some response $y \in \mathbb{R}^q$, also known as multi-task (or multi-output) regression (Zhang & Yang, 2018; Borchani et al., 2015; Xu et al., 2019). Thus, we might wish to construct a prediction set such that some some $q$-dimension version of $y$, say $y = (y^{(1)}, \ldots, y^{(q)})^\top$, is contained with some specified probability.

**Contributions**   With these potential scenarios is mind, we aim to extend exact and approximate conformal inference to the multi-task setting. Specifically, we contribute:

- extensions of exact conformal inference to multiple dimensions with various predictors and conformity measures

- `unionCP` to approximate conformal prediction sets without model retraining

- a multivariate extension of `rootCP` (Ndiaye & Takeuchi, 2021) which utilizes numerical root-based methods to find points on the boundary of a conformal prediction set

The introduction of `unionCP` and extension of `rootCP` provide a trade-off between various conformal inference methods, balancing the computational efficiency of split conformal prediction (`splitCP`) with the performance of full conformal prediction (`fullCP`). Table 1 summarizes the overall computational costs for each of these methods in terms of the number of model retraining iterations required to generate the conformal prediction region. We also include the computation complexity of the CP approximation provided by a grid-based approach (`gridCP`). In contrast, `fullCP` comprises approaches where the exact conformal prediction set can be constructed in a closed-form.

Table 1: Computational complexity of methods where $q$ is the response dimension, $m$ is the cardinality of the candidate value set, $d$ is the number of search directions, and $\epsilon$ is the tolerance.

| Method | Linear | Nonlinear |
|---|---|---|
| `splitCP` | $\mathcal{O}(q)$ | $\mathcal{O}(q)$ |
| `gridCP` | $\mathcal{O}(q)$ | $\mathcal{O}(mq)$ |
| `fullCP` | $\mathcal{O}(q)$ | - |
| `unionCP` | $\mathcal{O}(q)$ | $\mathcal{O}\big(ndq \log_2(1/\epsilon)\big)$ |
| `rootCP` | $\mathcal{O}(q)$ | $\mathcal{O}\big(dq \log_2(1/\epsilon)\big)$ |

From Table 1, we can see that in the linear case, each of the methods for prediction set generation require the same number of model refits as `splitCP`. We note that this does not account for the complexity of interval construction in each case.

The rest of the paper is laid out as follows. Section 2 provides requisite background for the paper. Section 3 extends exact conformal inference to multiple dimensions, while Section 4 introduces various conformal prediction set approximation methods in multiple dimensions. Section 5 provides empirical evaluation of our proposed approaches. Section 6 concludes the paper.

**Notation**   We denote the design matrix $X = (x_1, \ldots, x_n, x_{n+1})^\top$. Given $j \in [n]$, the rank of an element $u_j$ among a sequence $\{u_1, \ldots, u_n\}$ is defined as

$$\text{Rank}(u_j) = \sum_{i=1}^{n} \mathbb{1}_{u_i \leq u_j} \ .$$

## 2   Conformal Inference

In this section we provide background on relevant topics for this paper. Our applications within with paper are focused on regression, so we focus our background discussion on regression as well.

Originally introduced in Gammerman et al. (1998) as "transductive inference", conformal inference (CI) was originally focused on providing inference with classification approaches. Vovk et al. (2005) provides a formalized introduction to conformal inference within regression. With the express purpose of inference, the goal of CI is to attach, in some fashion, a measure of uncertainty to a predictor, specifically through the construction of a conservative prediction set, *i.e.,* one such that

$$\mathbb{P}\big(y_{n+1} \in \Gamma^{(\alpha)}(x_{n+1})\big) \geq 1 - \alpha. \tag{4}$$

We define $\mathcal{D}_n = \{(x_i, y_i)\}_{i=1}^{n}$ as a collection of $n$ observations, where the $i$-th data tuple $(x_i, y_i)$ is made up of a covariate vector $x_i$ and a response $y_i$. We wish to construct a *valid* prediction set for a new observation $(x_{n+1}, y_{n+1})$, where $x_{n+1}$ is some known covariate vector and $y_{n+1}$ is some, yet-to-be-observed response. Assuming each data pair $(x_i, y_i)$ and $(x_{n+1}, y_{n+1})$ are drawn exchangeably from some distribution $\mathcal{P}$, conformal inference generates conservative, finite-sample valid prediction sets in a distribution-free manner.

The main approach to perform the test inversion associated with Equation (2) relies on Lemma 1.

**Lemma 1.** *Let $U_1, \ldots, U_n, U_{n+1}$ be an exchangeable sequence of random variables. Then, for any $\alpha \in (0, 1)$,*

$$\mathbb{P}\big(\text{Rank}\,(U_{n+1}) \leq \lceil (1-\alpha)(n+1) \rceil\big) \geq 1 - \alpha.$$

In a prediction setting, test inversion for a particular candidate value $z$ is achieved by training the model of interest on an augmented data set $\mathcal{D}_{n+1}(z) = \mathcal{D}_n \cup (x_{n+1}, z)$. At this point, we leave our model of interest general, denoting the prediction of the $i$-th observation based on a model trained with $\mathcal{D}_{n+1}(z)$ as $\hat{y}_i(z)$. Following the refitting, each observation in the augmented data set receives a (non)conformity *measure*, which determines the level of (non)conformity between itself and other observations. One popular, and particularly effective, conformity measure is the absolute residual

$$S_i(z) = |y_i - \hat{y}_i(z)|. \tag{5}$$

We can construct the conformity *score* associated with a particular candidate point $z$ with

$$\pi(z) = \frac{1}{n+1} + \frac{1}{n+1} \sum_{i=1}^{n} \mathbb{1}_{S_i(z) \leq S_{n+1}(z)}, \tag{6}$$

where $S_i(z)$ is the conformity measure for the data pair $(x_i, y_i)$ as a function of $z$ and $S_{n+1}(z)$ is the conformity measure associated with $(x_{n+1}, z)$. Then, a valid $p$-value for the test shown in Equation (2) can be found with

$$p\text{-value}(z) = 1 - \pi(z).$$

A prediction set for an unknown response $y_{n+1}$ associated with some covariate vector $x_{n+1}$ is

$$\Gamma^{(\alpha)}(x_{n+1}) = \{z : (n+1)\pi(z) \leq \lceil (1-\alpha)(n+1) \rceil \}. \tag{7}$$

Then, by Lemma 1, with

$$(n+1)\pi(y_{n+1}) \equiv \text{Rank}(S_{n+1}(y_{n+1})),$$

Equation (4) holds for $\Gamma^{(\alpha)}(x_{n+1})$. By the previous results, CI can also be utilized in the multivariate response case, where one is interested in quantifying uncertainty with respect to the joint behavior of a collection of responses, given a set of covariates. Thus, we can construct a multidimensional prediction set $\Gamma^{(\alpha)}(x_{n+1}) \subset \mathbb{R}^q$ such that Equation (4) holds when $y_{n+1}$ is some $q$-dimensional random vector.

The first result extending conformal inference to the multivariate setting comes from Lei et al. (2015), which applies conformal inference to functional data, providing bounds associated with prediction "bands". Diquigiovanni et al. (2022) extends and generalizes additional results for conformal inference on functional data. Joint conformal prediction sets outside the functional data setting are explored in Kuleshov et al. (2018) and Neeven & Smirnov (2018). Messoudi et al. (2020; 2021) extend these works through the use of Bonferroni- and copula-based conformal inference, respectively. Cella & Martin (2020), Kuchibhotla (2020) and Johnstone & Cox (2021) construct joint conformal sets through the use of depth measures, *e.g.,* half-space and Mahalanobis depth, as the overall conformity measure. Applications of conformal inference have been seen in healthcare (Olsson et al., 2022), drug discovery (Cortés-Ciriano & Bender, 2019; Eklund et al., 2015; Alvarsson et al., 2021), and decision support (Wasilefsky et al., 2023), to name a few. For a thorough treatment on conformal inference in general, we point the interested reader to Fontana et al. (2023) and Angelopoulos et al. (2023).

## 2.1 Computationally Efficient Conformal Inference

Due to the inherent model refitting required to generate prediction sets through full conformal inference, *i.e.,* the testing of an infinite amount candidate points, more computationally efficient methods have been explored. We describe a subset of these methods in the following sections. Specifically, we focus on resampling-based and exact conformal inference.

### 2.1.1 Resampling Methods

Split conformal inference (Vovk et al., 2005; Lei et al., 2018) generates conservative prediction intervals under the same assumptions of exchangeability as *full* conformal inference . However, instead of refitting a model for each new candidate value, split conformal inference utilizes a randomly selected partition of $\mathcal{D}_n$, which includes a training set $\mathcal{I}_1$ and a calibration set $\mathcal{I}_2$. First, a prediction model fit using $\mathcal{I}_1$. Then, conformity measures are generated using out-of-sample predictions for observations in $\mathcal{I}_2$. The split conformal prediction interval for an incoming $(x_{n+1}, y_{n+1})$, when using the absolute residual as our conformity measure, is

$$\Gamma^{(\alpha)}_{\texttt{split}}(x_{n+1}) = [\hat{y}_{n+1} - s, \hat{y}_{n+1} + s], \tag{8}$$

where $\hat{y}_{n+1}$ is the prediction for $y_{n+1}$ generated using the observations in $\mathcal{I}_1$, and $s$ is the $\lceil (|\mathcal{I}_2|+1)(1-\alpha) \rceil$-th largest conformity measure for observations in $\mathcal{I}_2$. In order to combat the larger widths and high variance associated with split conformal intervals, cross-validation (CV) approaches to conformal inference have also been implemented. The first CV approach was introduced in Vovk (2015) as cross-conformal inference with the goal to "smooth" inductive conformity scores across multiple folds. Aggregated conformal predictors Carlsson et al. (2014) generalize cross-conformal predictors, constructing prediction intervals through any exhangeable resampling method, *e.g.,* bootstrap resampling. Other resampling-based conformal predictors also include CV+ and jackknife+ (Barber et al., 2021). For a more detailed review and empirical comparison of resampling-based conformal inference methods, we point the interested reader to Contarino et al. (2022).

### 2.1.2 Exact Conformal Inference for Piecewise Linear Estimators

In order to test a particular set of candidate values for inclusion in $\Gamma^{(\alpha)}(x_{n+1})$, we must compare the conformity measure associated with our candidate data point to the conformity measures of our training

data. Naively, this requires the refitting of our model for each new candidate value. However, Nouretdinov et al. (2001) showed that $S_i(z)$, constructed using Equation (5) in conjunction with a ridge regressor, varies piecewise-linearly as a function of the candidate value $z$, eliminating the need to test a dense set of candidate points through model refitting. Other exact conformal inference methods include conformal inference through homotopy (Lei, 2019; Ndiaye & Takeuchi, 2019), influence functions (Bhatt et al., 2021; Cherubin et al., 2021), and root-finding approaches (Ndiaye & Takeuchi, 2021). While not exact, Ndiaye (2022) provide approximations to the full conformal prediction region through stability-based approaches.

## 3 Exact Conformal Inference for Multi-task Learning

In the following sections, we extend the results in Nouretdinov et al. (2001) to multiple dimensions. We also discuss closed-form solutions for more general predictors as well as higher dimension prediction sets with other conformity measures. While CI can be applied to any prediction or classification task, in this section we restrict each of our predictors, given an incoming observation $(x_{n+1}, z)$, to the form

$$\hat{y}^{(k)}(z_k) = H_k(x_{n+1}, x_i)y^{(k)}(z_k), \tag{9}$$

where $\hat{y}^{(k)}(z_k)$ is the vector of predictions for the $k$-th response as a function of the candidate value $z_k$, and the candidate value *vector* is defined as $z = (z_1, \ldots, z_q)^\top$. We note that the restriction shown in Equation (9) is analogous to the restriction identifed in Equation (3). Additionally, we require that $H_k$ be constructed independently of $y^{(k)}$, *i.e.,* not as a function of $y^{(k)}$. Even with this restriction, $H_k$ is general enough so as to include many classes of predictors with examples described below. We specifically discuss how to construct exact $p$-values for a given $z$ without retraining our model. We also identify how we construct explicit *p-value change-point sets*, denoted as $\mathcal{E}_i$ for the $i$-th observation, where

$$\mathcal{E}_i \equiv \{z \in \mathbb{R}^q : S_{n+1}(z) \leq S_i(z)\}, \tag{10}$$

with the end goal of generating exact conformal prediction sets. Note that $\mathcal{E}_{n+1} \equiv \mathbb{R}^q$. Then, the $p$-value associated with the hypothesis test shown in Equation (2) for any candidate point $z$ is

$$p\text{-value}(z) = \frac{|\{i \in [n+1] : z \in \mathcal{E}_i\}|}{n+1}. \tag{11}$$

In the following sections, we describe several predictors with which we can perform exact conformal inference. We also describe exact conformal inference results for two conformity measure constructions, $\ell_1$ and $||\cdot||_W^2$, as well as results for finding points on the boundary of a conformal prediction set for any conformity measure.

### 3.1 Predictors For Exact Conformal Inference

Many regression methods generate predictions that follow Equation (9), including: ridge regression, kernel regression, and $k$-nearest neighbors, among others. In the sequel, we describe the predictions resulting from these approaches in the form laid out prior in Equation (3).

**Ridge Regression** While we have already described $H$ with respect to a ridge regressor using regularization parameter $\lambda$ in Section 1, we explicitly describe it for the $k$-th response dimension as

$$H_k(x_{n+1}) = X(X^\top X + \lambda_k I)^{-1}X^\top. \tag{12}$$

**Local Constant (Nadaraya-Watson) Regression** Kernel regression (Nadaraya, 1964; Watson, 1964) is a nonparametric regression technique that utilizes kernel density estimators (Parzen, 1962). Traditionally, a "kernel" is of the form

$$K_h(u) = \frac{1}{h}K\left(\frac{u}{h}\right), \tag{13}$$

where $K(\cdot)$ is a (symmetric) probability density, and $h$ is a bandwidth parameter. Often a Guassian kernel is used, *i.e.*, $K(u) \equiv \Phi(u)$, but other kernels are also popular. Using a kernel, we can perform nonparametric regression.

For some $x_i$ in $\mathcal{D}_{n+1}(z)$, the Nadaraya-Watson regression estimator generates a prediction

$$\hat{y}_i(z) = \sum_{j=1}^{n+1} \frac{K_h(x_i - x_j)}{\sum_{t=1}^{n+1} K_h(x_i - x_t)} y_j(z)$$

where $y_j(z)$ is the $j$-th element of $y(z)$. Thus, we can perform "local-constant" regression by using $H_k(x_{n+1}, x_i) \equiv H_k(x_{n+1}) = (w_1, \dots, w_{n+1})^\top$ where each $w_i$ is a vector of the normalized kernel values $K_h(\cdot)$ for each observation $x_j$ centered on $x_i$, *i.e.*,

$$w_i = \big(K_h(x_i - x_1), \dots, K_h(x_i - x_{n+1})\big)^\top \Big(I_{n+1} \sum_{j=1}^{n+1} K_h(x_i - x_j)\Big)^{-1}.$$

The current form for our kernel is general. However, we have not specified how it can be extended to a multivariate scenario; this is especially important for applications where we consider multiple covariates. While there do exist multivariate kernels that take vector arguments and utilize a bandwidth *matrix*, simpler approaches provide a more attractive and computationally efficient approach to generating multivariate kernels. As an example, a *product* kernel (Scott, 1992) generates a multivariate kernel by multiplying marginal kernel functions for each covariate. Namely, for an argument $u = (u_1, \dots, u_p)^\top$ and bandwidth vector $h = (h_1, \dots, h_p)^\top$,

$$K_h(u) = \prod_{k=1}^{p} K_{h_p}(u_p). \tag{14}$$

Another popular method for extension to multiple covariates are radial basis functions (Broomhead & Lowe, 1988), which utilize a norm as an argument to a univariate kernel.

**Local Linear Regression**   In contrast to the local-constant regression with the Nadaraya-Watson estimator, local-linear regression (Fan, 1992) utilizes a weighted version of the covariate matrix. Using the kernel introduced in Equation (13), we can construct a vector $w_i$ for the $i$-th observation. Local-linear regression then constructs an estimate for $y_i$, as a function of the candidate value pair $(x_{n+1}, z)$, by using an adjusted covariate matrix,

$$\tilde{X}_i = \begin{bmatrix} 1 & (x_1 - x_i)^\top \\ \vdots & \vdots \\ 1 & (x_{n+1} - x_i)^\top \end{bmatrix},$$

and a diagonalized version of $w_i$, which we identify as $G(x_i)$, resulting in predictions for the $i$-th observation of the $k$-th response of the form

$$\hat{y}_i^{(k)}(z_k) = \big(H_k(x_{n+1}, x_i) y^{(k)}(z_k)\big)_i,$$

where

$$H_k(x_{n+1}, x_i) = \tilde{X}_i \big(\tilde{X}_i^\top G(x_i) \tilde{X}_i\big)^{-1} \tilde{X}_i^\top G(x_i),$$

and $(\cdot)_i$ is the $i$-th element of the vector argument.

**$k$-nearest Neighbors**  $k$-Nearest neighbors (Cover & Hart, 1967; Fix, 1985) is a nonparametric regression technique that generates predictions based on neighboring values of an incoming observation. Traditionally, $k$ is used to describe the number of neighbors utilized to construct a prediction. In this work, we use $m$. Specifically, given a fixed $m$, with an observation for some $x$ we can construct a set of $m$ neighbors of $x$, made up of the training data observations. We define the set of neighbors for $x$ as $N(x)$. Then, a matrix $H_k(x_{n+1}, x_i) \equiv H_k(x_{n+1})$ can be constructed such that for each row-column position $(i, j)$,

$$H_k(x_{n+1})_{ij} = \begin{cases} 1/m & x_j \in N(x_i) \\ 0 & \text{otherwise} \end{cases}. \tag{15}$$

The result of Nouretdinov et al. (2001) was extended to include both lasso and elastic net regressors in Lei (2019). For this paper, we utilize a generalized version, shown in Proposition 1.

**Proposition 1.** *Assume the fitted model as in Equation (3), where $H(x_{n+1}, x_i) \equiv H$. Then, if we define $y(z) = (y^\top, z)^\top$, we can describe the vector residuals associated with the augmented dataset and some candidate value $z$ as*

$$\hat{y}(z) - Hy(z) = A - Bz$$

*where $A$ and $B$ are of the form*

$$A = (I - H)y(0)$$
$$B = (I - H)(0, \ldots, 0, 1)^\top.$$

*Proof.* By assumption, we can describe our vector of predictions $\hat{y}(z) = Hy(z)$. Thus,

$$\begin{aligned} y(z) - \hat{y}(z) &= y(z) - Hy(z) \\ &= (I_{n+1} - H)y(z) \\ &= (I_{n+1} - H)y(0) + (I_{n+1} - H)(0, \ldots, 0, 1)^\top z \\ &= A + Bz \end{aligned}$$

$\square$

With Proposition 1, we can then describe the conformity measure for the $i$-th observation, when using Equation (5), as $S_i(z) = |a_i + b_i z|$.

### 3.2 Exact $p$-values with $\ell_1$

We formalize our extension of Nouretdinov et al. (2001) to multiple dimensions, specifically utilizing

$$S_i(z) = ||y_i - \hat{y}_i(z)||_1, \tag{16}$$

as our conformity measure, in Proposition 2.

**Proposition 2.** *Assume the fitted model, $\hat{y}^{(k)}(z_k) = H_k(x_{n+1}, x_i)y^{(k)}(z_k)$. Then, using Equation (16),*

$$S_i(z) = ||a_i + b_i z||_1,$$

*where $a_i = (a_{1i}, \ldots, a_{qi})^\top$, $b_i = (b_{1i}, \ldots, b_{qi})^\top$, and $a_{ki}$ and $b_{ki}$ are the $i$-th elements of the vectors $A_k$ and $B_k$, respectively, defined as*

$$A_k = (I - H_k(x_{n+1}, x_i))y^{(k)}(0), \tag{17}$$

*and,*

$$B_k = (I - H_k(x_{n+1}, x_i))(0, \ldots, 0, 1)^\top. \tag{18}$$

---

**Algorithm 1** exact conformal prediction with $\ell_1$

---

**Input:** data $\mathcal{D}_n = \{(x_1, y_1), \ldots, (x_n, y_n)\}$, and $x_{n+1}$
Coverage level $\alpha \in (0, 1)$
Dimension $q$

    # *Initialization*

Construct $H_k(x_{n+1})$ for each $k = 1, \ldots, d$.
Construct $a_{ki}, b_{ki}$ for all $i = 1, \ldots, n + 1$.
Construct $\tilde{z}$ as in Equation (19).

    # *Construct p-value change-point sets*

We define $\tilde{z}_{(j)}$ as $\tilde{z}$ without the $j$-th component.
**for** $i \in 1, \ldots, n$ **do**
    Initialize set of corner points $\mathcal{V} = \emptyset$.
    **for** $j \in 1, \ldots, q$ **do**
        Fix $z_k = \tilde{z}_k$ for $k \neq j$. Then, find the set

$$z_j^* = \{z_j : c_i + |a_{ji} + b_{ji} z_j| = |a_{jn+1} + b_{jn+1} z_j|\}$$

    Set $\mathcal{V} = \mathcal{V} \cup \{(\tilde{z}_{(j)}, z_j^*)\}$
    **end for**
    Set $\mathcal{E}_i = \tilde{\mathrm{chull}}\{\mathcal{V}\}$ where $\tilde{\mathrm{chull}}\{S\}$ is the convex hull of the set $S$.
**end for**

**Return:** $\mathcal{E} = \{\mathcal{E}_i\}_{i=1}^n$

---

*Proof.* The proof follows directly from applying Proposition 1 to each element of the vector. $\qquad\square$

Proposition 2 allows us to construct conformity measures associated with a multidimensional response without retraining the model for each new $z$. Additionally, using Proposition 2 for each observation $(x_i, y_i)$, we can generate a region $\mathcal{E}_i$, as defined in Equation (10). Additionally, we can construct a fixed-point solution for $\hat{y}_{n+1}(z)$, *i.e.*, a point where $\hat{y}_{n+1}(z) = z$, as

$$\tilde{z} = \left( -\frac{a_{1n+1}}{b_{1n+1}}, \ldots, -\frac{a_{qn+1}}{b_{qn+1}} \right). \tag{19}$$

Equation (19) can be derived by setting each component of $S_{n+1}(z)$ equal to zero; the fixed point for a given observation is where the probability of a more extreme response, *i.e.*, $p$-value$(z)$, is maximized.

It is initially unclear *how* the construction of an individual region $\mathcal{E}_i$ occurs. As it stands, finding all $z$ such that $S_i(z) = S_{n+1}(z)$ is a multidimensional root-finding problem with infinite solutions, which has exponential complexity as $q$ increases. However, utilizing the inherent structure of each $\mathcal{E}_i$, we can simplify the problem. In order to provide clarity, we include Algorithm 1 to construct $\mathcal{E}_i$ in practice when using $\ell_1$.

Algorithm 1 leverages the fact that when using $\ell_1$, each $\mathcal{E}_i$ can be defined by the convex hull of a collection of points, specifically points axis-aligned with the fixed-point solution $\tilde{z}$. These points, referred to as "corners" within this paper, differ from $\tilde{z}$ in only the $j$-th element. The $j$-th element of a corner point, defined as $z_j^*$, is such that

$$z_j^* = \{z_j : c_i + |a_{ji} + b_{ji} z_j| = |a_{jn+1} + b_{jn+1} z_j|\}, \tag{20}$$

where $c_i = \sum_{k \neq j} |a_{ki} + b_{ki}\tilde{z}_k|$. We emphasize that all other components for the $n+1$-th conformity measure are zero by definition of $\tilde{z}$. Then, $z_j^*$ is either one of two solutions, $z_j^l$ or $z_j^u$. With the decomposition shown in Equation (20), finding $\mathcal{E}_i$ is reduced to a series of $q$ one-dimensional root-finding problems. We include a two-dimensional visual of the solutions generated using Algorithm 1 for an observation from the `cement` dataset in Figure 1. We also include further discussion on Algorithm 1 in Supplementary Materials

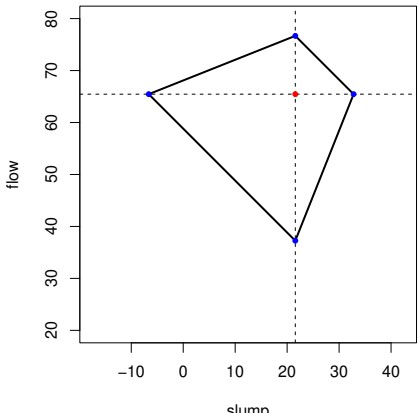

Figure 1: Example of Algorithm 1 for constructing $\mathcal{E}_i$ for an observation from `cement` dataset. The "●" identifies $\tilde{z}$, while the black line represents the border of the $p$-value change-point set The points $(\tilde{z}_1, z_2^l)$, $(\tilde{z}_1, z_2^u)$, $(z_1^l, \tilde{z}_2)$ and $(z_1^u, \tilde{z}_2)$ are identified with "●". The axes generated with $\tilde{z}$ are shown with the dotted black lines.

## 3.3 Exact $p$-values with $||\cdot||_W^2$

In order to generalize our exact $p$-value construction further than for use solely with $\ell_1$, we now consider conformity measures of the form

$$S_i(z) = r_i(z)^\top W r_i(z) \equiv ||r_i(z)||_W^2, \tag{21}$$

where $r_i(z) = y_i - \hat{y}_i(z)$, and $W$ is some $q \times q$ matrix. Proposition 3 provides a similar result to Proposition 2, but instead utilizes Equation (21). Namely, $S_i$ becomes quadratic with respect to $z$, instead of piecewise-linear.

**Proposition 3.** *Assume the fitted model $\hat{y}^{(k)}(z_k) = H_k(x_{n+1}, x_i)y^{(k)}(z_k)$ for each response dimension $k \in [q]$. Then, using Equation* (21),

$$S_i(z) = \begin{bmatrix} a_{1i} + b_{1i}z_1 \\ \vdots \\ a_{qi} + b_{qi}z_q \end{bmatrix}^\top W \begin{bmatrix} a_{1i} + b_{1i}z_1 \\ \vdots \\ a_{qi} + b_{qi}z_q \end{bmatrix}$$

*where $a_{ki}$ and $b_{ki}$ are the $i$-th elements of the vectors $A_k$ and $B_k$, respectively, as defined in Equation* (17) *and Equation* (18).

*Proof.* Let $S_i(z)$ be constructed as in Equation (21). Then, by Proposition 1, each element of the vector of residuals can be described in the form of $a_{1k} + b_{1k}z_k$, which gives us the desired result. □

With Proposition 3, the difference between $S_{n+1}(z)$ and $S_i(z)$ is the difference between two quadratic forms. Thus, we can describe the boundary $\mathcal{E}_i$ for every $i \in [n]$ as a *conic section*. Specifically, we can describe the difference between the candidate conformity measure and the conformity measure for observation $i$ as,

$$S_{n+1}(z) - S_i(z) = \begin{bmatrix} a_{1n+1} + b_{1n+1}z_1 \\ \vdots \\ a_{qn+1} + b_{qn+1}z_q \end{bmatrix}^\top W \begin{bmatrix} a_{1n+1} + b_{1n+1}z_1 \\ \vdots \\ a_{qn+1} + b_{qn+1}z_q \end{bmatrix} - \begin{bmatrix} a_{1i} + b_{1i}z_1 \\ \vdots \\ a_{qi} + b_{qi}z_q \end{bmatrix}^\top W \begin{bmatrix} a_{1i} + b_{1i}z_1 \\ \vdots \\ a_{qi} + b_{qi}z_q \end{bmatrix}. \quad (22)$$

Knowing we aim to find the boundary of each $\mathcal{E}_i$, *i.e.,* the roots of Equation (22), we can expand the statement into the form of a conic section such that

$$(1, z_1, \ldots, z_q)^\top [M_{n+1} - M_i](1, z_1, \ldots, z_q) = 0, \quad (23)$$

where $M_i$ is

$$M_i = \begin{bmatrix} \beta_{i0} & \beta_{i1}/2 & \cdots & \beta_{iq}/2 \\ \beta_{i1}/2 & \beta_{i11} & \cdots & \beta_{iq}/2 \\ \vdots & \vdots & \ddots & \vdots \\ \beta_{iq}/2 & \beta_{qi}/2 & \cdots & \beta_{iqq} \end{bmatrix}, \quad (24)$$

with the construction of each element of $M_i$ shown in Table 2.

Table 2: $\beta$ parameter formulas.

| Parameter | Formula |
|---|---|
| $\beta_{i0}$ | $\sum_{k=1}^{q}\sum_{j=1}^{q} a_{ik}a_{ij}w_{kj}$ |
| $\beta_{ik}$ | $2\sum_{j=1}^{q} a_{ij}b_{ik}w_{kj}$ |
| $\beta_{ikj}$ | $b_{ik}b_{ij}w_{kj}$ |

We can then translate a point $s$ on the unit-ball to the boundary of $\mathcal{E}_i$ with

$$z = \sqrt{K_i}L_i s + z_i^c$$

where $L_i$ is the upper-triangular Cholesky matrix of $M_i^* \equiv M_{n+1} - M_i$. We define $(\cdot)_{qq}$ as the lower $q \times q$ submatrix of the argument and $(\cdot)_{qqi}$ as the $i$-th row of the lower $q \times q$ submatrix. $z_i^c$ is the center of $\mathcal{E}_i$, *i.e.,*

$$z_i^c = (M_i^*)_{qq}^{-1}(-M_i^*)_{qq1} \quad (25)$$

and $K_i$ is

$$K_i = \frac{-det(M_i^*)}{det\big((M_i^*)_{qq}\big)}.$$

In order to maintain the probabalistic guarantees inherent to conformal inference, we require $W$ to be constructed exchangeably. Two constructions that satisfy exchangeability are: 1) $W$ constructed independently of $\mathcal{D}_{n+1}(z)$, or 2) $W$ constructed using all observations within $\mathcal{D}_{n+1}(z)$. However, we show in Section 5 that, in practice, setting $W \equiv \hat{\Sigma}^{-1}$, the observed inverse-covariance matrix associated with the residuals from our $q$ responses using a model constructed using only $\mathcal{D}_n$, performs well. The $p$-value associated with some $z$ using sets constructed using Equation (21) is the same as in Equation (11).

While Proposition 3 does not restrict the structure of $W$, limiting $W$ to be a symmetric, positive semi-definite matrix ensures that the set $\mathcal{E}_i$ is not only convex, but ellipsoidal. Without this additional restriction on the matrix $W$, the $p$-value change-point sets could be ill-formed, *i.e.,* non-convex. An example of an ill-formed $p$-value change-point set is shown in Figure 2.

For clarity, we include Algorithm 2 to describe how each $\mathcal{E}_i$ can be constructed in practice when using $|| \cdot ||_W^2$.

We also compare the exact $p$-value change-point sets constructed using $|| \cdot ||_W^2$ with $W = \hat{\Sigma}^{-1}$ to the conformal prediction $p$-value contours constructed using `gridCP` in Figure 3. Exact $p$-value change-points sets constructed using $\ell_1$ and $|| \cdot ||_W^2$ with $W = I_q$ are shown in Figure 4.

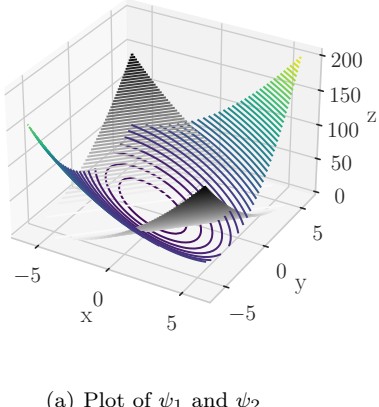
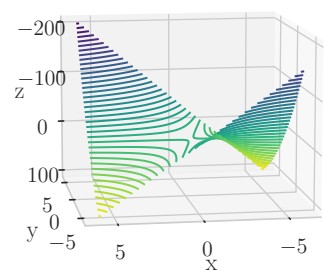

(a) Plot of $\psi_1$ and $\psi_2$

(b) Plot of the difference $\psi_1 - \psi_2$

Figure 2: $\psi_1 : z \mapsto \|y_0 - T_1 z\|$ and $\psi_2 : z \mapsto \|T_2 z\|$ where $y_0 = (1,0)^\top$, $T_1 = \left(\begin{smallmatrix} -1 & -1 \\ -1 & 0 \end{smallmatrix}\right)$ and $T_2 = \left(\begin{smallmatrix} 0 & -1 \\ 0 & 1 \end{smallmatrix}\right)$.

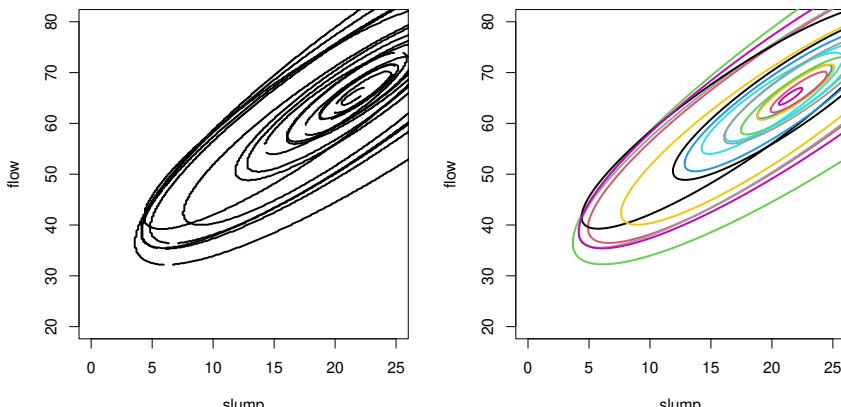

Figure 3: Comparing `gridCP` sets (left) to closed-form $p$-value change-point sets (right) constructed using $\|\cdot\|_W^2$ with $W = \hat{\Sigma}^{-1}$

.

---

**Algorithm 2** exact conformal prediction with $||\cdot||_w^2$

---

**Input:** data $\mathcal{D}_n = \{(x_1, y_1), \ldots, (x_n, y_n)\}$, and $x_{n+1}$
Coverage level $\alpha \in (0, 1)$
Dimension $q$
Matrix $W$ of dimension $q \times q$

    *# Initialization*

Construct $H_k(x_{n+1})$ for each $k = 1, \ldots, d$.
Construct $a_{ki}, b_{ki}$ for all $i = 1, \ldots, n+1$.

    *# Construct p-value change-point sets*

**for** $i \in 1, \ldots, n$ **do**
    Generate matrix $M_i^* \equiv M_{n+1} - M_i$ with $M_{n+1}$ and $M_i$ constructed as in Equation (24).
    Generate $\mathcal{E}_i = \{z : (1, z)_i^{*(1,z)\leq 0}\}$
**end for**

**Return:** $\mathcal{E} = \{\mathcal{E}_i\}_{i=1}^n$

---

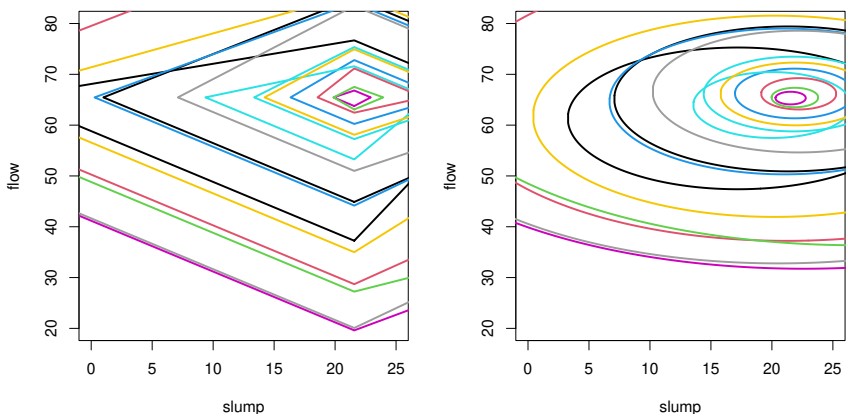

Figure 4: $p$-value change-point sets in two-dimensions for observations from single `cement` test data point with $\ell_1$ conformity measure constructed using Algorithm 1 (left), and $||\cdot||_W^2$ conformity measure with $W = I_2$ (middle) constructed using Algorithm 2.

## 3.4 Sampling points in the boundary

To cope with higher-dimensional complexity, we now aim to sample points on the boundary of the conformal prediction set. We can describe our results more generally by finding roots to $S_{n+1}(z) - S_i(z)$ where

$$z(t, d) = z_0 + td \tag{26}$$

for some direction vector $d \in \mathbb{R}^q$ and some interior point $z_0$. Then, finding roots is limited to finding $t^*$ such that

$$t^* = \{t > 0 : S_{n+1}(z(t, d)) - S_i(z(t, d)) = 0\}.$$

Restricting our models to linear predictors that follow Equation (3) in one direction of the space is equivalent to restricting the observed output. As such, we have

$$
\begin{aligned}
\hat{y}(z(t,d) = Hy(z(t,d)) &= H(y_1, \ldots, y_n, z(t,d))^\top \\
&= Hy(0) + H(0, \ldots, 0, z_0 + td)^\top \\
&= Hy(z_0) + tH(0, \ldots, 0, d)^\top
\end{aligned}
$$

The conformity measures along the direction $d$ are then given by

$$
\begin{aligned}
S_i(z(t,d)) &= \|y_i - \hat{y}_i(z(t,d))\| = \|a_i - tb_i\| \\
S_{n+1}(z(t,d)) &= \|z(t,d) - \hat{y}_{n+1}(z(t,d))\| = \|a_{n+1} - tb_{n+1}\|,
\end{aligned}
$$

where we define

$$
\begin{aligned}
a_i &= y_i - (Hy(z_0))_i, & a_{n+1} &= z_0 - (Hy(z_0))_{n+1} \\
b_i &= H_{n+1}d, & b_{n+1} &= d - H_{n+1}d
\end{aligned}
$$

The goal is now to solve the one dimension problem $\psi(t) = S_{n+1}(z(t,d)) - S_i(z(t,d)) \geq 0$. Without this one dimensional restriction, the computations are significantly more difficult and impossible to track without stronger data assumptions. This is illustrated in Figure 2 where we provide simple examples that lead to a non-convex set of solutions. For completeness, we describe below the solution for different norms to be used as score functions and explicit form of the conformal set for a given direction.

**Solving for $\ell_1$ norm**   We have

$$
\begin{aligned}
\psi(t) &= \|a_{n+1} - tb_{n+1}\|_1 - \|a_i - tb_i\|_1 \\
&= \langle \text{sign}(a_{n+1} - tb_{n+1}), a_{n+1} - tb_{n+1} \rangle - \langle \text{sign}(a_i - tb_i), a_i - tb_i \rangle \\
&= c(t) + ts(t)
\end{aligned}
$$

where we can easily see that $\psi$ is piecewise linear with slopes $s(t)$ and bais $c(t)$ defined as

$$
\begin{aligned}
s(t) &= \langle \text{sign}(a_i - tb_i), b_i \rangle - \langle \text{sign}(a_{n+1} - tb_{n+1}), b_{n+1} \rangle \\
c(t) &= \langle \text{sign}(a_{n+1} - tb_{n+1}), a_{n+1} \rangle - \langle \text{sign}(a_i - tb_i), a_i \rangle
\end{aligned}
$$

Every piece is characterized by the moment where the sign terms change *i.e.,* when for a coordinate $j \in [q]$, it holds

$$
a_{i,j} - tb_{i,j} = 0 \text{ or } a_{n+1,j} - tb_{n+1,j} = 0 \tag{27}
$$

Without loss of generality, let us assume that $b_{i,j}$ and $b_{n+1,j}$ are non zero; otherwise, the equation does not have a solution and we can skip them. Also let us assume that $a_{i,j}$ and $a_{n+1,j}$ are non zero, otherwise the solution is trivial equal to zero. As such, we have $2q$ solutions of Equation (27) that we denote

$$
t_1^\star, \ldots, t_{2q}^\star
$$

By the Intermediate Value Theorem, the function $t \mapsto \psi(t)$ is equal to zero if and only if it exists consecutive roots $t_k^\star, t_{k+1}^\star$ for which $\psi$ takes opposite sign *i.e.,* $\psi(t_k^\star)\psi(t_{k+1}^\star) \leq 0$ (note that the product is equal to zero only at the roots). Then, we deduce that $\mathcal{E}_i \equiv \{z \in \mathbb{R}^q : S_{n+1}(z) \leq S_i(z)\}$ restricted on the line $z_0 + td$ is a union of intervals whose boundaries are delimited by the roots of $\psi$ that are easily obtained as

$$
\hat{t}_k = -\frac{c(t_{k+1}^\star)}{s(t_{k+1}^\star)} \text{ or } \hat{t}_k = -\frac{c(t_k^\star)}{s(t_k^\star)}.
$$

**Solving for general Mahalanobis score function**   By monotonicity, squaring the norm in the definition of $\psi$ does not change its level set. So, let's consider the function

$$
\begin{aligned}
\psi(t) &= \|a_{n+1} - tb_{n+1}\|_M^2 - \|a_i - tb_i\|_M^2 \\
&= t^2(\|b_{n+1}\|_M^2 - \|b_i\|_M^2) - 2t(\langle a_{n+1}, b_{n+1} \rangle_M - \langle a_i, b_i \rangle_M) + (\|a_{n+1}\|_M^2 - \|a_i\|_M^2)
\end{aligned}
$$

where we denote $\|\alpha\|_M^2 = \langle \alpha, \alpha \rangle_M$ and $\langle \alpha, \beta \rangle_M = \langle \alpha, M\beta \rangle$. Hence the root of $\psi$ are obtained by merely solving a quadratic equation.

**Explicit description of the conformal set** For any $i$ in $[n+1]$, we denote the intersection points of the functions $S_i(z_t)$ and $S_{n+1}(z_t)$ by $t_i$s and then we have $\mathcal{E}_i$ can be an interval (possibly a point), a union of interval or even empty. In all cases, it is characterized by the intersection points obtained explicitly. Since $\pi(z(t,d))$ is piecewise constant, it changes only at those points. We denote the set of solutions $t_1, \cdots, t_K$ in increasing order as $t_0 < t_1 < \cdots < t_K$. Whence for any $t$, it exists a unique index $j = \mathcal{J}(t)$ such that $t \in (t_j, t_{j+1})$ or $t \in \{t_j, t_{j+1}\}$ and for any $t$, we have

$$(n+1)\pi(z_t) = \sum_{i=1}^{n+1} \mathbb{1}_{t \in S_i} = N(\mathcal{J}(t)) + M(\mathcal{J}(t))$$

where the functions

$$N(j) = \sum_{i=1}^{n+1} \mathbb{1}_{(t_j, t_{j+1}) \subset S_i} \text{ and } M(j) = \sum_{i=1}^{n+1} \mathbb{1}_{t_j \in S_i}$$

Note that $\mathcal{J}^{-1}(j) = (t_j, t_{j+1})$ or $\mathcal{J}^{-1}(j) = \{t_j, t_{j+1}\}$. Finally, we have the restriction of the conformal set to the direction $d$ is given by

$$\Gamma^{(\alpha)}(x_{n+1}, d) = \bigcup_{\substack{j \in [K] \\ N(j) > (n+1)\alpha}} (t_j, t_{j+1}) \ \cup \bigcup_{\substack{j \in [K] \\ M(j) > (n+1)\alpha}} \{t_j\} \ . \tag{28}$$

## 4 Approximate Conformal Inference for Multi-task Learning

While the results in Section 3 allows for the construction of exact $p$-values with no additional model refitting (for multiple responses), we still cannot describe exactly the conformal prediction sets in closed-form. Thus, we aim to construct approximations of the conformal prediction set for a given $x_{n+1}$. In this section we specifically introduce a union-based approximation for a conformal prediction set generated using the results from Section 2.1.2. Additionally, we extend the root-based approximation procedures introduced in Ndiaye & Takeuchi (2021) to the multi-task setting.

### 4.1 `unionCP` Approximation Method

After constructing the set $\mathcal{E}$ for an incoming point $x_{n+1}$, it is initially unclear which regions $\mathcal{E}_i$ make up various conformal prediction sets, let alone how we need to combine these regions to get the exact conformal prediction sets. Thus, we aim to provide an approximation of conformal prediction sets using the regions generated with the approaches introduced in Section 3. We provide Proposition 4 to bound error probabilities associated with potential combinations of these regions.

**Proposition 4.** *Under uniqueness of conformity measures, for some* $y \in \mathbb{R}^q$ *such that* $(x_1, y_1), \ldots, (x_{n+1}, y_{n+1})$ *are drawn exchangeably from* $\mathcal{P}$*, for any* $\mathcal{S} \subset [n]$

$$\mathbb{P}\left(y \in \bigcup_{i \in \mathcal{S}} \mathcal{E}_i\right) \geq \frac{|\mathcal{S}|}{n+1}.$$

*Proof.* Assume we have the data pair $(x, y)$ drawn exchangeably with $(x_1, y_1), \ldots, (x_n, y_n)$. Also assume that we have constructed the set $\mathcal{E}$. In Section 3, we show construction of $\mathcal{E}$ with $\ell_1$ and $|| \cdot ||_W^2$ as conformity measures, but the following proof holds for any conformity measure. First, we select and fix any $\mathcal{S} \subseteq [n]$. We then fix a $z$ such that $z \notin \bigcup_{i \in \mathcal{S}} \mathcal{E}_i$. Then,

$$z \notin \bigcup_{i \in \mathcal{S}} \mathcal{E}_i \iff S_i(z) \leq S_{n+1}(z) \ \forall \ i \in \mathcal{S} \implies \sum_{i=1}^{n+1} \mathbb{1}\{S_i(z) \leq S_{n+1}(z)\} \geq |\mathcal{S}| + 1$$

Then, for $y$

$$\mathbb{P}\Big(y \notin \bigcup_{i \in \mathcal{S}} \mathcal{E}_i\Big) \leq \mathbb{P}\Big(\sum_{i=1}^{n+1} \mathbb{1}\{S_i(y) \leq S_{n+1}(y)\} \geq |\mathcal{S}| + 1\Big)$$

$$\Rightarrow \mathbb{P}\Big(y \in \bigcup_{i \in \mathcal{S}} \mathcal{E}_i\Big) \geq 1 - \mathbb{P}\Big(\sum_{i=1}^{n+1} \mathbb{1}\{S_i(y) \leq S_{n+1}(y)\} \geq |\mathcal{S}| + 1\Big)$$

$$\geq \mathbb{P}\Big(\sum_{i=1}^{n+1} \mathbb{1}\{S_i(y) \leq S_{n+1}(y)\} \leq |\mathcal{S}|\Big).$$

By Lemma 1, with the selection of $\alpha = 1 - \frac{|\mathcal{S}|}{n+1}$,

$$\mathbb{P}\Big(\sum_{i=1}^{n+1} \mathbb{1}\{S_i(y) \leq S_{n+1}(y)\} \leq |\mathcal{S}|\Big) \geq \frac{|\mathcal{S}|}{n+1}.$$

$\square$

Proposition 4 states that with the selection of *any* subset of $\mathcal{E}$, the probability of the response $y_{n+1}$ being contained in the union of that subset is bounded-below by a function of cardinality. For example, if we wish to construct, say, a conservative 50% prediction set, we could select (at random) a set $\mathcal{S} \subseteq \mathcal{E}$ such that $|\mathcal{S}| \geq |\mathcal{E}|/2$; the union of all sets within $\mathcal{S}$ would provide a conservative prediction set. We again note that while our work emphasizes $\ell_1$ and $||\cdot||_W^2$, Proposition 4 holds for any conformity measure.

Now, the random set constructed might not provide tight coverage as there exist some $\mathcal{E}_i$ such that

$$\bigcup_{i' \in \mathcal{S}_{(i)}} \mathcal{E}_{i'} \subseteq \mathcal{E}_i,$$

where $\mathcal{S}_{(i)}$ is some subset of $[n]$ that does not contain $i$; some $p$-value change-point sets are contained in others and, thus, choosing the larger set could result in extremely conservative coverage. We include results related to the theoretical coverage associated with the randomized approach in Supplemental Materials.

While the union of a random selection of regions forms a conservative $1 - |\mathcal{S}|/(n+1)$ prediction set, we can provide more intelligently constructed sets that are empirically less conservative (but still valid). Suppose we provide an ordering of our regions, where $\mathcal{E}_{(k)}$ is defined as the $k$-th smallest region by volume.

**Definition 1** (unionCP). *A more efficient $(1-\alpha)$ prediction set approximation can then be constructed as*

$$\hat{\Gamma}^{(\alpha)}(x_{n+1}) = \bigcup_{i \in \mathcal{S}_{1-\alpha}} \mathcal{E}_{(i)}, \tag{29}$$

*where $\mathcal{S}_{1-\alpha} = [\lceil(1-\alpha)(n+1)\rceil]$. We dub the approximation shown in Equation (29) as* unionCP.

By Proposition 4, unionCP generates an approximation that, at minimum, provides a region that is at least valid. We compare prediction sets constructed using unionCP to a random selection of regions for multiple predictors in Section 5. We find empirically that sets constructed using unionCP are less conservative than a random collection of $p$-value change-point sets.

While Proposition 4 and the adjustment described in Equation (29) allow for conservative prediction sets, at times, the union of various $\mathcal{E}_i$ does not explicitly describe a conformal prediction set exactly. Thus, unionCP provides (at worse) a conservative approximation of the true conformal prediction set. Figure 5 provides an example where the region constructed with unionCP differs from the true conformal prediction set.

With full comformal prediction, the computational complexity depends heavily on the number of candidate values chosen, while the computational burden of unionCP depends on the number of observations $n$. To reduce the computation required to generate the approximation, we can utilize the result shown in Lemma 2.

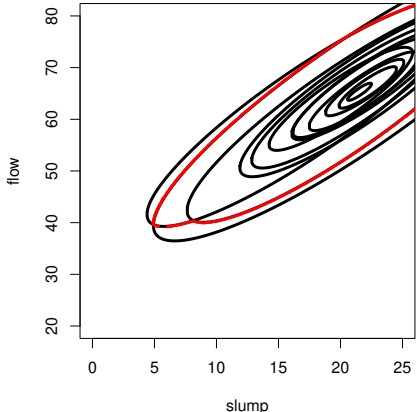

Figure 5: Comparison of full conformal prediction set for $\alpha = .25$ (red line) and regions included in `unionCP` approximation (black line(s)).

---

**Algorithm 3** `unionCP`

---

**Input:** data $\mathcal{D}_n = \{(x_1, y_1), \ldots, (x_n, y_n)\}$, and $x_{n+1}$
Coverage level $\alpha \in (0, 1)$
Subset size $m \leq n$

    *# Initialization*

Generate a random subset $M \subseteq [n]$ of size $m$.
Construct $\mathcal{E}^m = \{\mathcal{E}_i\}_{i \in M}$ with Algorithm 1 or Algorithm 2.

    *# Construct conformal prediction region approximation*

Order each element of $\mathcal{E}^m$ by volume where $\mathcal{E}^m_{(k)}$ is defined as the $k$-th smallest region by volume of $\mathcal{E}^m$.
Generate $\mathcal{S}_{1-\alpha} = \{1, \ldots, \lceil (1-\alpha)(m+1) \rceil \}$.
Set $\hat{\Gamma}^{(\alpha)}(x_{n+1}) = \bigcup_{i \in \mathcal{S}_{1-\alpha}} \mathcal{E}^m_{(i)}$.

**Return:** $\hat{\Gamma}^{(\alpha)}(x_{n+1})$

---

**Lemma 2.** *Let $U_1, \ldots, U_n, U_{n+1}$ be an exchangeable sequence of random variables. Then, any subsample $U_1, \ldots, U_m$ is also exchangeable.*

By Lemma 2, we can randomly select any $m$ observations, where $1 < m \leq n$, and the conformity measures of this subset, along with $S_{n+1}(z)$, will also be exchangeable. Thus, we can randomly select a subset of $\mathcal{E}$ of size $m$, defined as $\mathcal{E}^m$, and then order this subset by volume, where $\mathcal{E}^m_{(k)}$ is defined as the $k$-th smallest region by volume of the set $\mathcal{E}^m$. Then, by Proposition 4, `unionCP` constructed with this subset also provides valid prediction regions, at a potentially much lower computational cost.

If we wish to avoid the `unionCP` approximation, we can generate exact $p$-values using Equation (11) in conjunction with a grid-based approach with much computational gain over that of full conformal prediction. We include Algorithm 3 to construct a generalized version of the `unionCP` approximation of the conformal prediction set for a given test observation.

### 4.2 Connection between `unionCP` and `splitCP`

In this section, we provide further theoretical backing for `unionCP` by connecting it explicitly to `splitCP`. First, we can further generalize the conformal prediction set generated when using `splitCP` than just with the use of the absolute residual. In general, for an incoming $x_{n+1}$, the split conformal prediction set is

$$\Gamma_{\texttt{split}}^{(\alpha)}(x_{n+1}) = \{z : S_{n+1}(z) \le s\}, \tag{30}$$

where $S_{n+1}(z)$ is constructed as a function of $z$ and $\hat{y}_{n+1}$, generated using observations in $\mathcal{I}_1$, and $s$ is the $\lceil(|\mathcal{I}_2|+1)(1-\alpha)\rceil$-th largest conformity measure for observations in $\mathcal{I}_2$. In one dimension, it is easy to show that with the absolute residual $|z - \hat{y}_{n+1}|$, Equation (30) reduces to the region shown in Equation (8). For $y \in \mathbb{R}^q$, when using $||\cdot||_W^2$ as the conformity measure,

$$S_{n+1}(z) \le s \Rightarrow ||z - \hat{y}_{n+1}||_W^2 \le s$$

We also note that for `splitCP`, we can construct the $p$-value change-point set for observation $i$ when using $||\cdot||_W^2$ as

$$\mathcal{E}_i \equiv \{z : S_{n+1}(z) \le S_i(z)\} = \{z : ||z - \hat{y}_{n+1}||_W^2 \le ||y_i - \hat{y}_i||_W^2\}.$$

Now, if we select two observations $i$ and $j$ such that $S_i(z) \le S_j(z)$ then the result in Proposition 5 holds.

**Proposition 5.** *For two observations $i$ and $j$ such that $S_i(z) \le S_j(z) \,\forall\, z$,*

$$S_i(z) \le S_j(z) \Rightarrow \mathcal{E}_i \subseteq \mathcal{E}_j$$

*Proof.* We assume $S_i(z) \le S_j(z) \,\forall\, z$. We note this this assumption is valid for `splitCP` as the conformity measure for each observation is constant with respect to $z$. Thus, we can provide an ordering of the conformity measures. Then,

$$\begin{aligned}
\mathcal{E}_j &= \{z : S_{n+1}(z) \le S_j(z)\} \\
&= \{z : S_{n+1}(z) \le S_j(z) + S_i(z) - S_i(z)\} \\
&= \{z : S_{n+1}(z) \le S_i(z) + \underbrace{S_j(z) - S_i(z)}_{\ge 0 \text{ by assumption}}\} \\
&= \mathcal{E}_i \cup \{z : S_i(z) \le z \le S_j(z)\}
\end{aligned}$$

$\square$

Thus, with the `unionCP` approach, we can match exactly the conformal prediction sets constructed using `splitCP`. We note that this result is related to the *nested* conformal prediction sets discussed in Gupta et al. (2022).

### 4.3 Root-based Approximation Methods

As noted earlier, computation of the conformal prediction sets requires model readjustment for any candidate value to replace the true $y_{n+1}$ value. Current efficient approaches to exact computation, limited to dimension one, are restricted to models that are piecewise-linear; this structure allows to track changes in the conformity function. We have extended these approaches to higher dimensions in the previous section.

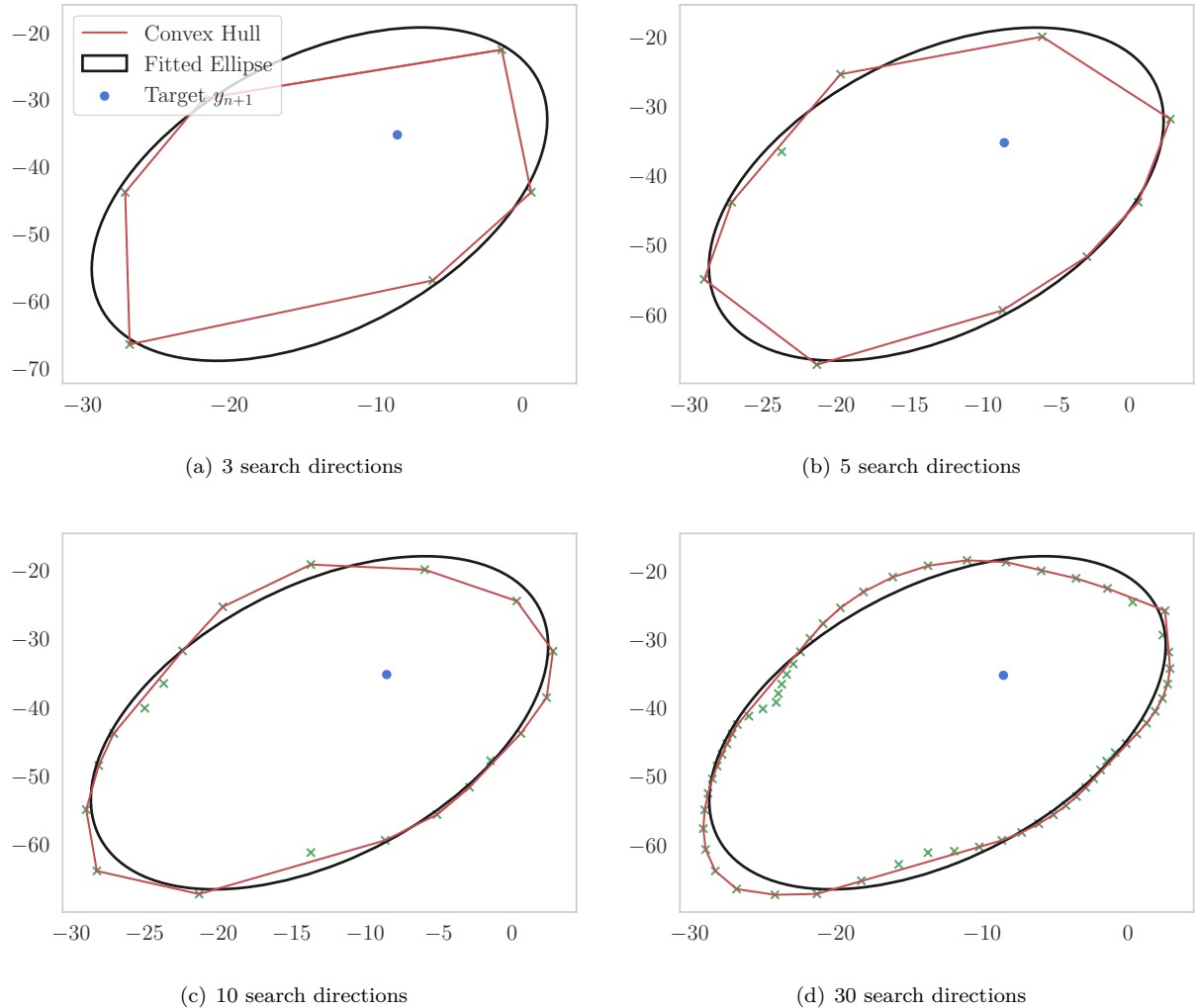

(a) 3 search directions

(b) 5 search directions

(c) 10 search directions

(d) 30 search directions

Figure 6: Illustration of the approximated conformal prediction set obtained fitting ellipse and convex hull given boundary points obtained by `rootCP`. We use scikit-learn `make_regression` to generate synthetic dataset with the parameters n_samples = 15, n_features = 5, n_targets = 2 is the dimension of in output $y_{n+1}$. We selected 80% of informative features and 60% for effective rank (described as the approximate number of singular vectors required to explain most of the input data by linear combinations) and the standard deviation of the random noise is set to 5.

To go beyond linear structures, we can use approximate homotopy approaches which, given an optimization tolerance, provide a discretization of all the values that $y_{n+1}$ can take. However, these approaches are also limited in dimension one and have an exponential complexity in the dimension of $y_{n+1}$. Convexity assumptions are also required, which, unfortunately, are not verified for more complex prediction models.

In this section, we extend the approximations of conformal prediction in multiple dimensions by computing conformal prediction set boundaries directly. Unlike the one-dimensional case where the boundary is often two points, in multiple dimensions the boundary is continuous and, thus, uncountable, which makes finite-time computation impossible.

To get around this difficulty, the main idea here is very simple. We will first fix a finite set of search directions; we will estimate the intersection points between the boundary of the conformal prediction set and the chosen direction. Then, we use the points on the boundary as a data base to fit a convex approximation, *e.g.,* an

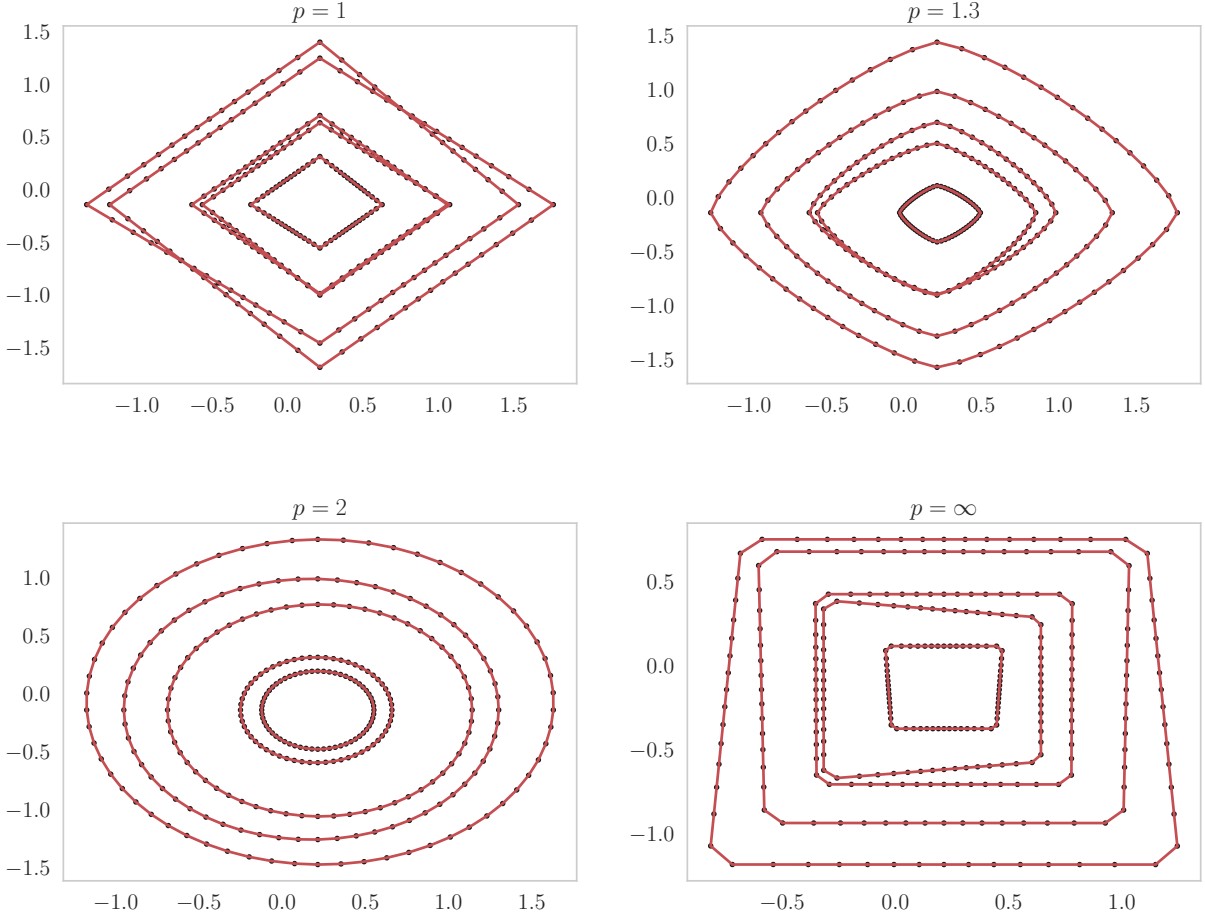

Figure 7: Illustration of the approximated $\mathcal{E}_i$ with 30 search directions with conformity measures defined with $\ell_p$ norms. Note that the different level sets can actually overlap. Solid black lines denote ellipsoid approximations of $\mathcal{E}_i$ using calculated boundary points.

ellipse or the convex hull, passing through these points. More formally, we want to estimate (efficiently) the set described in Equation (7).

**Assumptions.** We suppose that the conformal prediction set is *star-shaped i.e.,* there exists a point $z_0$ such that any other point $z$ within $\Gamma^{(\alpha)}(x_{n+1})$ can be connected to $z_0$ with a line segment.

Note that a star-shaped set can be non-convex; we can still approximate complex, e.g., non-convex, conformal sets. We provide some illustration in Figure 6. We also note that ellipsoidal sets (or any convex set) are inherently star-shaped.

**Outline of `rootCP`**

Given any direction $d \in \mathbb{R}^q$, the intersection points between the boundary of $\Gamma^{(\alpha)}(x_{n+1})$, defined as $\partial\Gamma^{(\alpha)}(x_{n+1})$, and the line passing through $z_0$ and directed by $d$ are obtained by solving the one dimensional equation

$$\pi(z(t,d)) = 1 - \alpha, \tag{31}$$

where $z(t,d)$ is as in Equation (26). We briefly described the main steps and display the detail in Algorithm 4.

1. Fit a model $\mu_0$ on the observed training set $\mathcal{D}_n$ and predict a feasible point $z_0 = \mu_0(x_{n+1})$.

2. For a collection of search directions $\{d_1, \ldots, d_K\}$, perform a bisection search in $[t_{\min}, 0]$ and $[0, t_{\max}]$ to output solutions $\hat{\ell}(d_k)$ and $\hat{u}(d_k)$ of Equation (31) at direction $d_k$, after at most $\log_2(\frac{t_{\max}-t_{\min}}{\epsilon_r})$ iterations for an optimization tolerance $\epsilon_r > 0$. Notice that the star-shape assumption implies that we will have only two roots on the selected directions.

3. Fit a convex set on the roots obtained at the previous step $\{\hat{\ell}(d_k), \hat{u}(d_k)\}_{k \in [K]}$. In practice, when one uses a least-squares ellipse as the convex approximation, a number of search directions $K$ proportional to the dimension $q$ of the target $y_{n+1}$ is sufficient. This is not necessarily the case for the convex hull. We refer to Figure 6 where we observe that many more search directions are needed to cover the conformal set when using the convex hull approximation.

---

**Algorithm 4** `rootCP` in multiple dimensions

**Input:** data $\mathcal{D}_n = \{(x_1, y_1), \ldots, (x_n, y_n)\}$, and $x_{n+1}$
Coverage level $\alpha \in (0, 1)$, accuracy $\epsilon_r > 0$, list of search directions $d_1, \ldots, d_K$

    *# Initialization*
Set $z_0 = \mu_0(x_{n+1})$ where we fitted a model $\mu_0$ on the initial training dataset $\mathcal{D}_n$

    *# Approximation of boundary points*
**for** $k \in \{1, \ldots, K\}$ **do**
  We define the direction-wise conformity function as

$$\pi_k(t) = \pi(z(t)) - \alpha \text{ where } z(t) = z_0 + td_k$$

    1.  $t^- = \texttt{bisection\_search}(\pi_k, t_{\min}, 0, \epsilon_r)$

    2.  $t^+ = \texttt{bisection\_search}(\pi_k, 0, t_{\max}, \epsilon_r)$

  Set $\hat{\ell}_{d_k} = z_0 + t^- d_k$ and $\hat{u}_{d_k} = z_0 + t^+ d_k$
**end for**

    *# Convex approximation*

$$\hat{\Gamma}(x_{n+1}) = \tilde{\text{co}}\left\{\hat{\ell}_{d_1}, \hat{u}_{d_1}, \ldots, \hat{\ell}_{d_K}, \hat{u}_{d_K}\right\}$$

where $\tilde{\text{co}}\{S\}$ is a convex set built on $S$, *e.g.,* the convex hull or fit an ellipse using the points in $S$
**Return:** $\hat{\Gamma}(x_{n+1})$

---

The root-finding approach can also be adapted to `unionCP` by approximating the level-line boundary of the $\mathcal{E}_i$ score difference introduced in Equation (10). In so doing, the previous restriction to quadratic functions that enabled an explicit construction is no longer necessary, at the cost of an approximation. We illustrate this generalization to different score functions in Figure 7.

## 5  Empirical Results and Application

To provide empirical support for our theoretical results, we consider a small data example using the multi-task `cement` data set (Yeh, 2007). For the sake of simplicity, we limit our exploration to two dimensions, focusing on the construction of prediction sets for slump and flow, given information on other elements in the mixture, *e.g.,* amount of cement, fly ash and superplasticizer.

We include empirical coverage results for the approximation approaches described in Section 4, specifically with regions constructed using $|| \cdot ||_W^2$. We also include results for the root-based approximation results

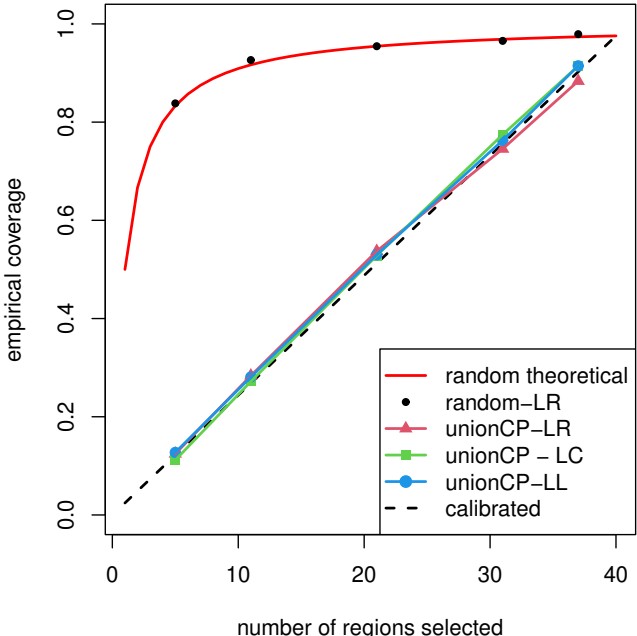

Figure 8: Comparison of empirical coverage with random selection of $k$ regions and `unionCP` for various predictors, including: linear regression (LR), Nadaraya-Watson (LC), and local-linear (LL) across 100 repetitions. The calibrated curve is constructed using *number of regions selected*$/(m+1)$, where $m = n = 40$.

described in Section 4.3 for a wide-array of predictors that include those more general than the restrictions we outline in our paper.

As a reference benchmark, from Lemma 1, we have $\pi(y_{n+1}) \geq \alpha$ with probability larger than $1 - \alpha$. Hence, we can define the `oracleCP` as $\pi^{-1}([\alpha, +\infty))$ where $\pi$ is obtained with a model fit optimized on the oracle data $\mathcal{D}_{n+1}(y_{n+1})$ on top of the root-based approach to find boundary points. We remind the reader that the target variable $y_{n+1}$ is not available in practice.

### 5.1 `unionCP` Approximation Application

While our focus for the construction of $H(x_{n+1}, x_i)$ has been general, for much of our discussion we often reference $H(\cdot)$ constructions associated with a ridge-regressor. Thus, in this section, we wish to demonstrate the performance of `unionCP` with other methods that fall under our the general model restriction; we specifically explore the prediction methods discussed in Section 3.1. We also note that there are additional methods other than these which also adhere to our model restrictions.

Figure 3 includes a comparison of the $p$-value change-point regions constructed with Equation (21) to the conformal prediction sets constructed using the grid-based approach, also with linear regression for each predictor as well as coverage results for various predictors on the cement data set. From Figure 8, we can see that the `unionCP` approximation, with each of the models, is empirically calibrated.

### 5.2 `rootCP` Approximation Application

We numerically examine the performance of `rootCP` on multi-task regression problems using both synthetic and real databases. The experiments were conducted with a coverage level of 0.9, *i.e.,* $\alpha = 0.1$.

For comparisons, we run the evaluations on 100 repetitions of examples, and display the average of the following performance statistics for different methods: 1) the empirical coverage, *i.e.,* the percentage of times

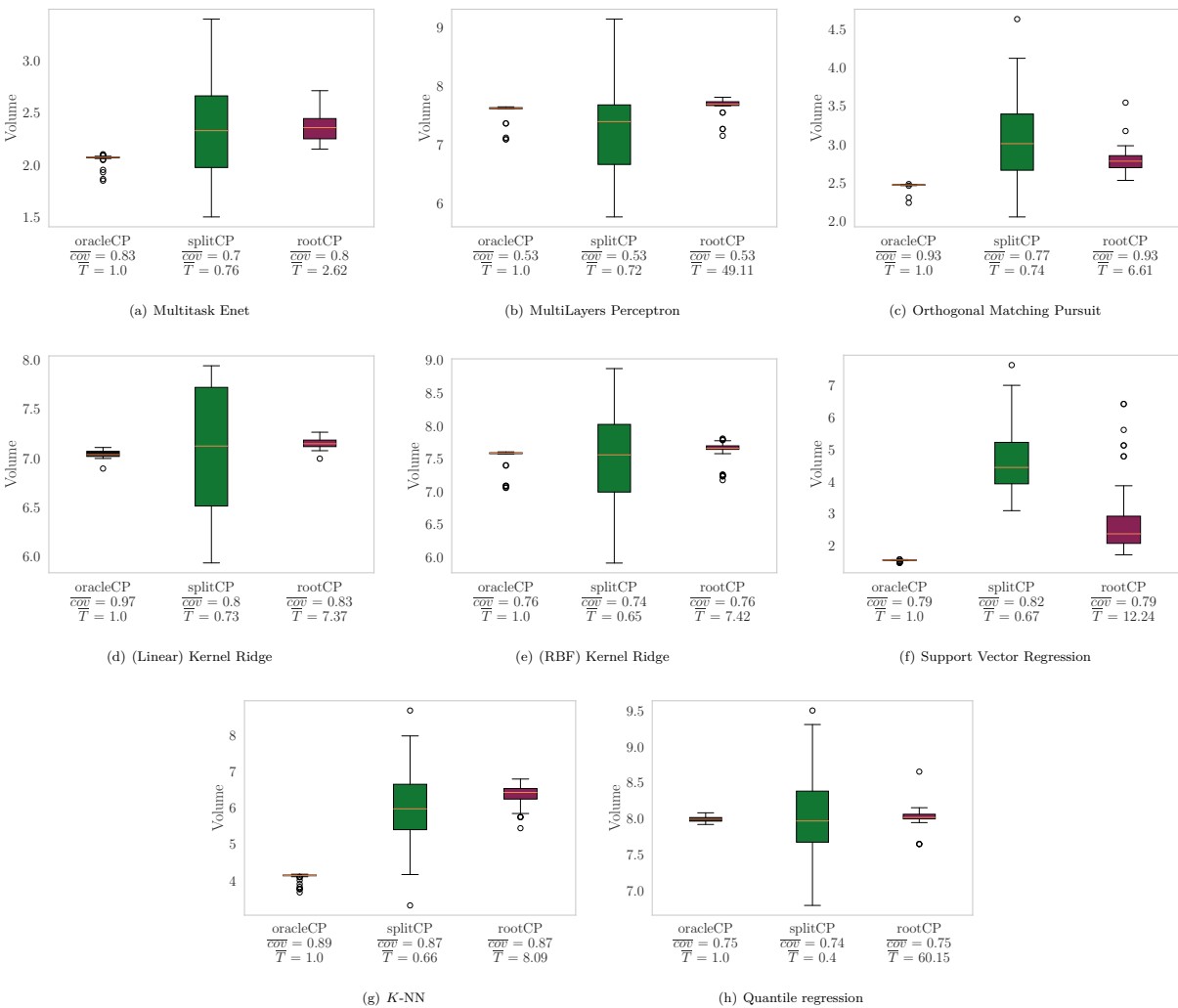

Figure 9: Ellipse Benchmarking conformal sets for several regression models on `cement` dataset. We display the volumes of the confidence sets over 100 random permutations of the data. We denoted $\overline{cov}$ the average coverage, and $\overline{T}$ the average computational time normalized with the average time for computing `oracleCP` which requires a single model fit on the whole data.

the prediction set contains the held-out target $y_{n+1}$, 2) the volume of the confidence intervals, and 3) the execution time. For each run, we randomly select a data tuple $(x_i, y_i)$ to constitute the targeted variables for which we will compute the conformal prediction set. The rest is considered as observed data $\mathcal{D}_n$. Similar experimental settings are considered in Lei (2019).

We run experiments on a suite of complex regression models, including: multi-task elastic net, multi-layer perceptron, orthogonal matching pursuit, kernel ridge regression with both linear and Gaussian kernels, support vector regression, $k$-nearest neighbors and quantile regression. The results are shown in Figure 9. We include additional results in Supplementary Materials.

## 6 Conclusion

In this paper, we introduced exact $p$-values in multiple dimensions for predictors that are a linear function of the candidate value. Specifically, we discussed the exact construction of $p$-values using various conformity measures, including $\ell_1$ and $||\cdot||_W^2$. Additionally, we introduced methods for various approximations of multidimensional

$1 - \alpha$ conformal prediction sets through union-based and root-based prediction set construction, `unionCP` and and a multi-task extension to `rootCP`, respectively. We also also deliver probabilistic bounds and convergence results for these approximations. We then showed empirically with multiple predictors, including a subset of both linear and nonlinear predictors, that these approximations were comparable to `tgridCP` sets, while drastically reducing the computational requirements.

One drawback of the methods described in this work is their lack of adaptability. We hope to include region adaptability in future work, e.g., with methods similar to those introduced in Messoudi et al. (2022).

Other questions about the theoretical guarantees of our approach have yet to be answered. For example, we lack precise characterizations on the number of points to be sampled on the conformal set boundary, as well as implications our convex approximations, *e.g.,* ellipse, convex hull, related to expected volume and potential coverage loss in the worst case. Besides the conformal sets presented in this paper, these questions are equally relevant to the construction of any high-dimensional confidence sets.

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

## Supplementary Materials

In the following sections, we include additional content to further support the contributions of the paper.

### S.1  Additional Experiments

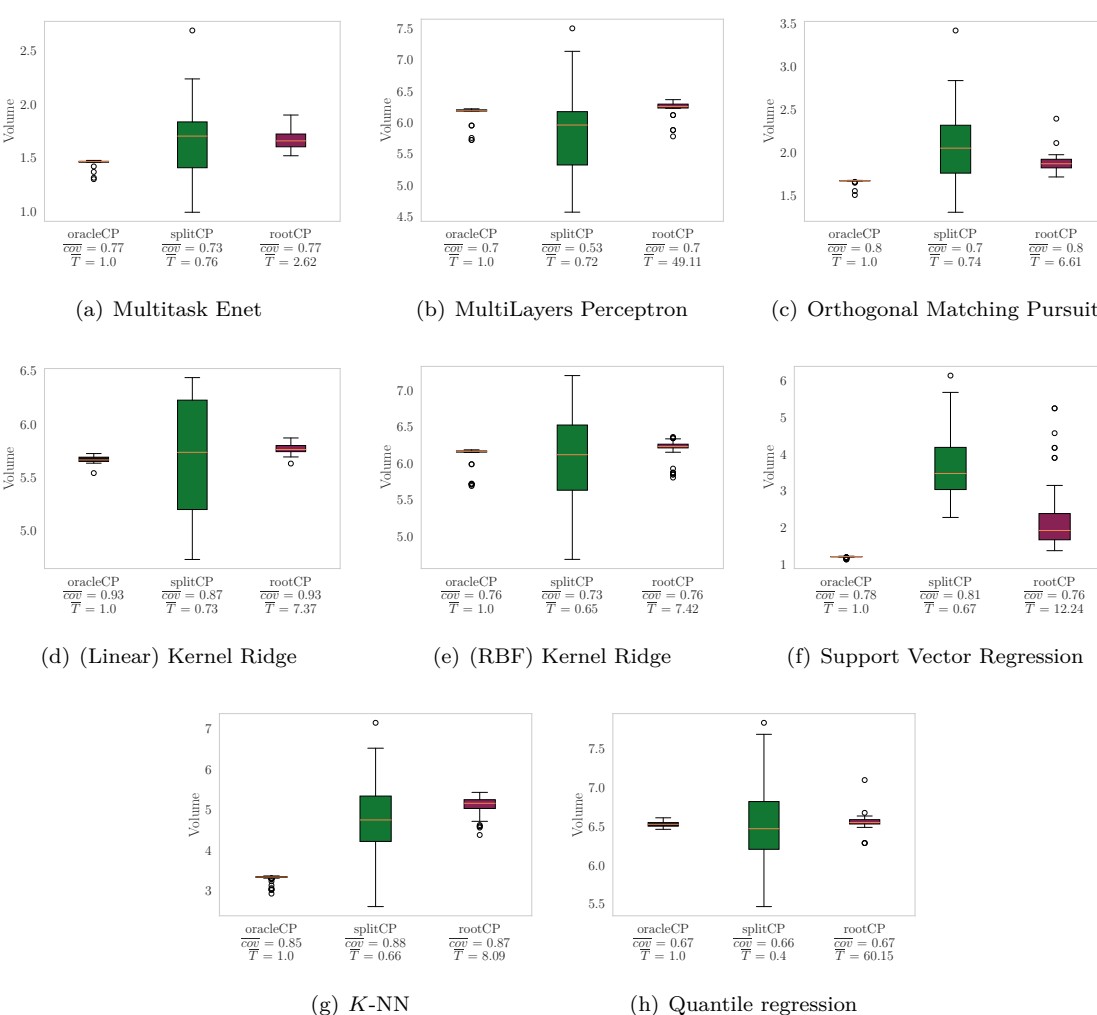

Figure S1: Benchmarking the convex Hull based conformal sets for several regression models on `cement` dataset. We display the lengths of the confidence sets over 100 random permutation of the data. We denoted $\overline{cov}$ the average coverage, and $\overline{T}$ the average computational time normalized with the average time for computing `oracleCP` which requires a single model fit on the whole data.

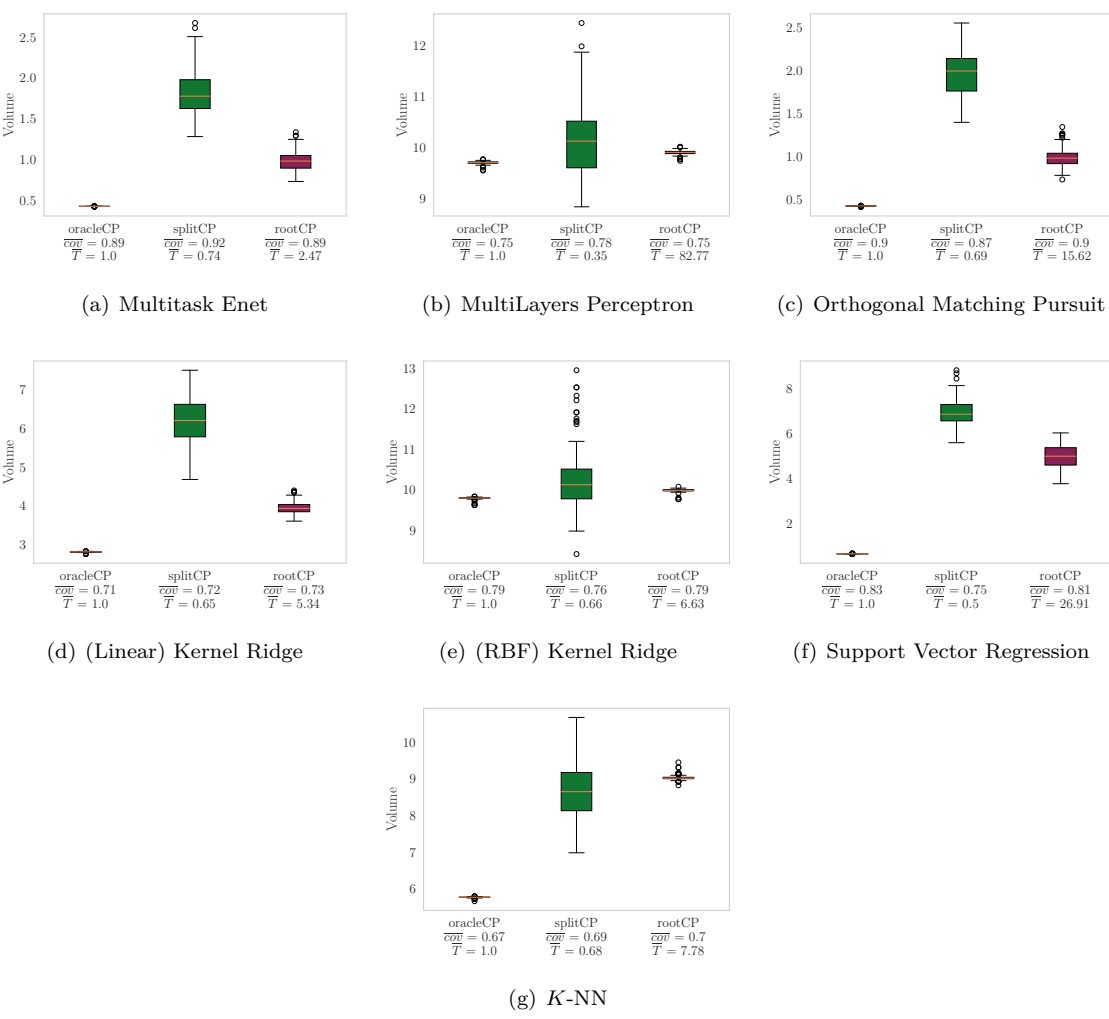

Figure S2: Benchmarking the ellipse based conformal sets for several regression models on `synthetic` dataset. We display the lengths of the confidence sets over 100 random permutation of the data. We denoted $\overline{cov}$ the average coverage, and $\overline{T}$ the average computational time normalized with the average time for computing `oracleCP` which requires a single model fit on the whole data.

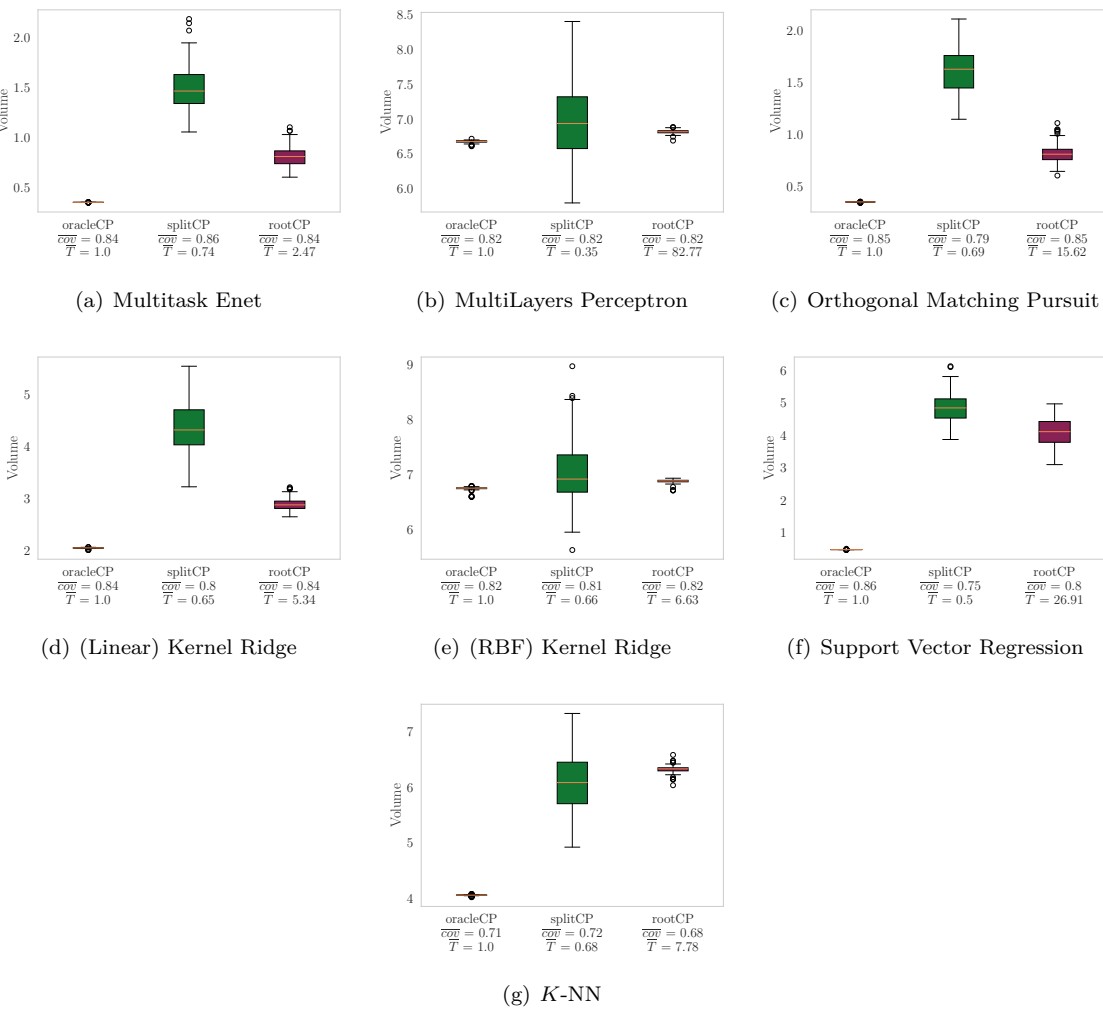

Figure S3: Benchmarking the convex Hull based conformal sets for several regression models on `synthetic` dataset. We display the lengths of the confidence sets over 100 random permutation of the data. We denoted $\overline{cov}$ the average coverage, and $\overline{T}$ the average computational time normalized with the average time for computing `oracleCP` which requires a single model fit on the whole data.

### S.2 Sketch-Proof of Random Region Selection Coverage Probability

We now include a sketch-proof for the probability that a randomly selected subset of $\mathcal{E}$, of cardinality $k$, constructed with $\mathcal{D}_{n+1}(z)$ will contain $y_{n+1}$.

*Proof.* First, we define $\Gamma_k$ as a random subset of $\mathcal{E}$ such that $|\Gamma_k| = k$ where each region $i$, constructed as a function of $\mathcal{D}_{n+1}(z)$, is selected with probability $1/n$. We assume, without loss of generality, that $\mathcal{E}_{(i)} \subset \mathcal{E}_{(i+1)} \forall i = 1, \ldots, n$. While this assumption does not always hold, it results in a larger upper bound. Then, for $y_{n+1}$,

$$\mathbb{P}\big(y_{n+1} \in \mathcal{E}_{(k)}\big) = \frac{k}{n+1} \tag{32}$$

Furthermore, let the random variable $e_i = 1$ if region $\mathcal{E}_{(i)}$ is selected (with probability $1/n$). Then, for $k = 1$,

$$
\begin{aligned}
\mathbb{P}(y_{n+1} \in \Gamma_1) &= \sum_{i=1}^{n} \mathbb{P}(y_{n+1} \in \Gamma_1, e_i = 1) \\
&= \sum_{i=1}^{n} \mathbb{P}(y_{n+1} \in \Gamma_1 | e_i = 1)\mathbb{P}(e_i = 1) \\
&= \frac{1}{n} \sum_{i=1}^{n} \mathbb{P}(y_{n+1} \in \Gamma_1 | e_i = 1) \\
&= \frac{1}{n} \sum_{i=1}^{n} \frac{i}{n+1} \\
&\quad \big(\text{by Equation (32)}\big) \\
&= \frac{1}{n(n+1)} \sum_{i=1}^{n} i \\
&= \frac{1}{n(n+1)} \frac{n(n+1)}{2} \\
&= \frac{1}{2}
\end{aligned}
$$

Now consider $\Gamma_k$ for $k > 1$.

We define the random variable $\pi_k$ as a joint random variable for $(e_1, \ldots, e_n)$ where the support of $\pi_k$ is $\Pi_k$, the set of all $2^n$ combinations of $e_i \in \{0, 1\}$. Because we limit our regions to be nested, the $k$-th region is contained in $(k+1)$-th region. Thus, we can describe coverage probabilities with various $\Gamma_k$ and $\pi_k$ by just

examining the highest $k$ such that $e_k = 1$, defined as $\max_k(\pi_k)$. Then,

$$
\begin{aligned}
\mathbb{P}(y_{n+1} \in \Gamma_k) &= \sum_{\pi_k \in \Pi_k} \mathbb{P}(y_{n+1} \in \Gamma_k, \pi_k) \\
&= \sum_{\pi_k \in \Pi_k} \mathbb{P}(y_{n+1} \in \Gamma_k | \pi_k) \mathbb{P}(\pi_k) \\
&= \sum_{i=k}^{n} \mathbb{P}(y_{n+1} \in \Gamma_k | \max_k(\pi_k) = i) \mathbb{P}(\max_k(\pi_k) = i) \\
&\quad \left( \mathbb{P}(\max_k(\pi_k) = i) = 0 \ \forall \ i < k \right) \\
&\geq \sum_{i=k}^{n} \frac{i}{n+1} \mathbb{P}(\max_k(\pi_k) = i) \\
&\quad \text{(by Lemma 1)} \\
&= \frac{1}{n+1} \sum_{i=k}^{n} i \mathbb{P}(\max_k(\pi_k) = i)
\end{aligned}
$$

where

$$
\mathbb{P}\left(\max_k(\pi_k) = i\right) = \frac{\binom{i-1}{k-1}}{\binom{n}{k}}.
$$

$\square$

### S.3 Reproducibility

The source code utilized for our experimentation will be available in open-source later.

### S.4 Discussion on Algorithm 1

There are several *potential* solutions to Equation (20); some of these solutions are invalid. In order to solve for valid solutions, we can find intervals where the argument of each absolute value component is less than zero. Without loss of generality, with two-dimensions,

$$
a_{2i} + b_{2i} z_2 < 0 \quad \Rightarrow \quad z_2 < -\frac{a_{2i}}{b_{2i}} \text{ and } a_{2n+1} + b_{2n+1} z_2 < 0 \Rightarrow \quad z_2 < -\frac{a_{2n+1}}{b_{2n+1}}. \tag{33}
$$

Then, we can construct a set of intervals

$$
(-\infty, l_i), (l_i, u_i), (u_i, \infty)
$$

to check for solutions, where $l_i = \min(-\frac{a_{2i}}{b_{2i}}, -\frac{a_{2n+1}}{b_{2n+1}})$ and $u_i = \max(-\frac{a_{2i}}{b_{2i}}, -\frac{a_{2n+1}}{b_{2n+1}})$. The left-most interval corresponds to both the arguments within Equation (20) being negative; the right-most interval corresponds to both the arguments within Equation (20) being positive, resulting in

$$
z_2(i) = \frac{c_1 + (-a_{2i} + a_{2n+1})}{b_{2i} - b_{2n+1}} \text{ and } z_2(i) = \frac{c_1 + (a_{2i} - a_{2n+1})}{b_{2n+1} - b_{2i}},
$$

respectively. We explicitly denote $z_2(i)$ as a function of the index $i$ because we must repeat this process for each observation. For simplicity, we drop the $i$ index. The sign of the components when $z_2$ is contained within the inner interval depends on the value of $l_i$ and $u_i$ where

$$l_i = -\frac{a_{2i}}{b_{2i}} \Rightarrow \text{ left component is positive, right component is negative.}$$

$$\text{otherwise} \Rightarrow \text{ right component is positive, left component is negative.}$$

Regardless, we can find the two valid solutions by checking all four potential solutions to see if they fall within their respective intervals. We denote these two valid solutions $z_2^l$ and $z_2^u$, corresponding to the smallest and largest solution value, respectively. We can repeat the above process, instead fixing $z_2 = \tilde{z}_2$ to find equivalent solutions for $z_1$, denoted $z_2^l$ and $z_2^u$. The points $(\tilde{z}_1, z_2^l)$, $(\tilde{z}_1, z_2^u)$, $(z_1^l, \tilde{z}_2)$ and $(z_1^u, \tilde{z}_2)$, all generated as a function of $i$, make up the "corners" of each region $\mathcal{E}_i$.

